# Decellularized Skin Extracellular Matrix (dsECM) Improves the Physical and Biological Properties of Fibrinogen Hydrogel for Skin Bioprinting Applications

**DOI:** 10.3390/nano10081484

**Published:** 2020-07-29

**Authors:** Adam M Jorgensen, Zishuai Chou, Gregory Gillispie, Sang Jin Lee, James J Yoo, Shay Soker, Anthony Atala

**Affiliations:** Wake Forest Institute for Regenerative Medicine, Wake Forest School of Medicine, Medical Center Boulevard, Winston-Salem, NC 27157, USA; zchou@wakehealth.edu (Z.C.); ggillisp@wakehealth.edu (G.G.); sjlee@wakehealth.edu (S.J.L.); jyoo@wakehealth.edu (J.J.Y.); ssoker@wakehealth.edu (S.S.)

**Keywords:** decellularization, skin anatomy and microarchitecture, wound healing, printability, rheology

## Abstract

Full-thickness skin wounds are a significant clinical burden in the United States. Skin bioprinting is a relatively new technology that is under investigation as a new treatment for full-thickness injuries, and development of hydrogels with strong physical and biological characteristics are required to improve both structural integrity of the printed constructs while allowing for a more normal extracellular matrix milieu. This project aims to evaluate the physical and biological characteristics of fibrinogen hydrogel supplemented with decellularized human skin-derived extracellular matrix (dsECM). The hybrid hydrogel improves the cell viability and structural strength of bioprinted skin constructs. Scanning electron microscopy demonstrates that the hybrid hydrogel is composed of both swelling bundles interlocked in a fibrin network, similar to healthy human skin. This hybrid hydrogel has improved rheological properties and shear thinning properties. Extrusion-based printing of the fibrinogen hydrogel + dsECM demonstrates significant improvement in crosshatch pore size. These findings suggest that incorporating the properties of dsECM and fibrinogen hydrogels will improve in vivo integration of the bioprinted skin constructs and support of healthy skin wound regeneration.

## 1. Introduction

Burn injuries are a significant clinical burden in the United States, with 1.1 million injuries annually requiring medical attention and an estimated cost of around $2 billion per year [1,2]. Autologous skin grafting is the current standard of care for burn wounds, but this is limited by sufficient amounts of harvest sites and can be scarce in patients with severe wounds [3]. Bioprinting is under investigation as a complementary method for in vitro fabrication of full-thickness skin with multiple cell types organized into biomimetic layers [4].

In both bioengineered and native tissues, extracellular matrix (ECM) provides cells with mechanical and structural support and gives cues for cell proliferation, migration and adhesion [5]. The ECM is composed of growth factors, structural and regulatory proteins, and signaling cascades, which are cannot be fully recapitulated in synthetic biomaterials [6]. Thus tissue decellularization can be used to collect ECM to be used as a biomaterial for tissue engineering.

Many methods have been examined for performing decellularization, including physical, chemical and enzymatic approaches, and generally multiple methods are combined for effective decellularization [7]. Skin tends to be more difficult to decellularize than other tissues due to its dense, complex structure and abundant lipid content [8]. In the decellularization process, trypsin is used as an enzymatic agent, cleaving peptide bonds on the C-side of Arg and Lys, effectively removing cellular components from ECM. Alcohols are then used to induce cell lysis by dehydration and solubilizing lipids. EDTA is used as a chelating agent, disrupting cells adhesion to ECM by binding to metallic cations [9,10]. Chemical detergents, including TritonX-100, are then used to solubilize cell membranes and nuclear components [7,11].

Decellularized extracellular matrix (dECM) retains ECM structural and functional properties such as nanostructure, biochemical complexity, and bio inductive properties. By removing its cellular and nuclear components the risk inflammatory and immune response is reduced. dECM has been shown to promote the in vivo creation of site-specific, functional tissue [8,9,12], making dECM highly desirable for tissue regeneration and wound healing applications. dECM has previously been processed into many different delivery forms, including powders, coatings, injectable hydrogels and hydrogels [8], and have been isolated from multiple organ systems, including skin, heart, liver, nerves, tendon, and blood vessel have been used in various tissue engineering applications [8]. While collagen is the major component of human skin ECM, other human skin ECM proteins may foster additional cellular attachment and activities [13]. Mammalian ECM has been commonly used as surgical mesh materials and as scaffolds for regenerative medicine applications, suggesting the ability to support cell growth [14].

In prior bioprinting studies, we have demonstrated that human skin constructs bioprinted with a fibrinogen hydrogel integrate, form an extracellular matrix, and remain viable when implanted onto athymic mice [4]. Figure 1A demonstrates that bioprinted skin constructs harvested 42 days post-implantation in mice have an immature extracellular matrix (ECM), whereas normal human dermis has a rich collagen ECM. This lack of ECM maturity may result from the relatively insufficient ECM components in the fibrinogen hydrogel used for bioprinting. It is also known that fibrinogen post-printing mechanical strength is limited [15], and the addition of collagen is known to increase fibrinogen hydrogel’s mechanical strength significantly [16]. The objective of this study is to determine if the addition of a decellularized skin ECM (dsECM) improves the biological and physical properties of fibrinogen hydrogel for skin bioprinting applications.

## 2. Materials and Methods

### 2.1. Decellularization and ECM Solubilization

Full-thickness skin tissues were obtained from discarded tissue under the Wake Forest University Health Sciences IRB protocol. Skin decellularization and ECM solubilization protocol were modified from a previous study [17,18]. After removing subcutaneous fat, connective tissues and hair, skin samples treated with the following solutions under constant agitation on an orbital shaker at 225 RPM in room temperature: 0.25% trypsin for 6 h, three 15 min washes of deionized water, 70% ethanol for 10 h, 3% hydrogen peroxide for 15 min, two 15 min washes of deionized water, 1% Triton X-100 in 0.26% EDTA and 0.69% Tris base for 6 h, then replace with fresh solution for 16 h, three 15 min washes of deionized water, 0.1% peracetic acid in 4% ethanol for 2 h, two 15 min washes of PBS, and finally two 15 min washes of deionized water. Treated samples were then frozen at −80 °C in 50 mL conical tubes for 24 h, lyophilized for 48 h, and powdered by a cryomill. After sterilization by gamma irritation, powdered ECM was enzymatically was digested in a sterile solution of 1:10 porcine pepsin: ECM by weight in 0.01 N HCl for 72 h on a shaker at room temperature. 8% (80 mg/mL), 4% (40 mg/mL), 3% (30 mg/mL), and 2% (20 mg/mL) of solubilized ECM were made separately. The supernatant was obtained after centrifuging at 4000 RPM for 10 min and then neutralized on ice with sterilized 1N NaOH to around pH 7.4.

### 2.2. Preparation of Fibrinogen Hydrogel and Fibrinogen + dsECM Hydrogel

Fibrinogen hydrogel was made based on a previous protocol [4]. Briefly, 100 μL/mL glycerol, 35 mg/mL gelatin from bovine skin, and 3 mg/mL hyaluronic acid were dissolved in Dulbecco’s modified Eagle’s medium (DMEM) overnight at 37 °C. 30 mg/mL fibrinogen from bovine plasma was added to mix for an additional 2 h. For fibrinogen + dsECM hydrogel, 2× fibrinogen hydrogel was made and then mixed with solubilized ECM solution at a 1:1 volumetric ratio (e.g., fibrinogen + 1.5% dsECM hydrogel = 2× fibrinogen hydrogel + 3% ECM). 

### 2.3. Cell Culture and Preparation of Cellularized Constructs

Primary human skin fibroblasts were isolated from full-thickness skin tissues obtained from discarded tissue with consent under Wake Forest University Health Sciences IRB protocol. The cells were cultured in 2D culture media (DMEM supplemented with 10% fetal bovine serum (FBS) and 1% penicillin/streptomycin [P/S]). 10 × 10^6^ cells/mL Fibroblasts were encapsulated in hydrogels with 40 μL/mL aprotinin. They were injected into 3D printed disc molds and crosslinked with 20 μL/mL thrombin at room temperature for 1 h. After incubating in 3D culture media (2D culture media with 20 μL/mL aprotinin) overnight, samples were transferred from disc molds into 16-well plates with 3D culture media. The media was changed every two days. Three biological replicates were made for each condition. 

### 2.4. Histology

Histology was performed to evaluate healthy human skin and bioprinted skin (Figure 1A), and cell encapsulation in fibrinogen only and fibrinogen + dsECM (Figure 1C). Briefly, samples were fixed in 4% paraformaldehyde for 48 h. Upon paraffin processing, a microtome (Leica) was used to generate 5 µm sections containing both the center and edge portions of the treated wounds. Slides were then stained with hematoxylin and eosin and Masson’s trichrome, and imaged by light microscopy. Images were then examined to determine the degree of dermal organization, keratin staining, collagen staining, extracellular matrix composition and organization, and the structural integrity of cellularized constructs (Figure 2). 

### 2.5. Cell Viability Assay

Live/dead cell viability assay was performed on cellularized constructs over 15 days. The samples were stained with 2 μM calcein AM and 4 μM EthD-1 for 1 h and imaged by confocal microscopy. Percent viability was determined using Image J. 

### 2.6. Rheological Characterization

Sample preparation: uncrosslinked samples were directly loaded onto the rheometer. Crosslinked samples were prepared by loading hydrogels on 3D printed disc molds with 12 mm diameter and 1 mm depth. Samples were cross-linked with 20 μL/mL thrombin in PBS solution for 1 h and immersed in DMEM overnight in cell culture condition. Then crosslinked disc samples were retrieved for rheometry.

pH effect on solubilized ECM: The consistency and viscosity of solubilized ECM pre (pH~3) and post (pH~7) neutralization were compared by visual inspection and a viscosity-temperature ramp test. Samples were loaded onto a 40 mm cone-plate geometry with a 500 μm gap. The temperature was ramped from 10 °C to 40 °C at the rate of 10 °C/min. An oscillatory strain of 0.4% was applied at a frequency of 1.0 Hz. Then the average viscosity between 15 °C and 25 °C was calculated for each sample.

Flow property: A frequency sweep test was conducted using a 40 mm cone-plate geometry with a 500 μm gap. An oscillatory strain of 0.1% was applied with an increasing shear rate from 0.01 to 100 Hz at 22 °C. Viscosity was plotted against shear rate. 

Temperature dependency: Gel properties over temperature were characterized by a temperature ramp test using a 12 mm parallel plate geometry (40 mm cone-plate geometry for viscosity measurement) with a 500 μm gap. The temperature was ramped from 10 °C to 40 °C at the rate of 10 °C/min. An oscillatory strain of 0.4% was applied at a frequency of 1.0 Hz. Storage modulus (G’), viscosity, and tan δ (G”/G’) was plotted against temperature. 

Thrombin-Fibrinogen crosslinking effect: A strain sweep test was performed on hydrogels before and after crosslinked by thrombin. Samples were loaded onto a 12 mm parallel plate geometry with a 500 μm gap. Increasing oscillatory strain from 0.02% to 2% was applied at 1.0 Hz at 22 °C. G’ were plotted against strain and the averages were calculated. 

Constructs mechanical strength: A strain sweep test (same parameters as the crosslinking effect test) was conducted on cell-laden constructs and gel-only constructs three times over 15 days to determine their mechanical strength. 

### 2.7. Scanning Electron Microscopy 

The human skin samples, ECM, and fibrinogen + dsECM were examined by scanning electron microscopy (SEM). The samples were dried by lyophilization. 8% solubilized ECM, fibrinogen hydrogel, and fibrinogen + 1.5% dsECM hydrogel were fixed in 2.5% glutaraldehyde for 24 h, followed by three 30 min PBS washes. The samples were then dehydrated in a series of alcohol (30, 50, 70, 90, 100% ethanol in PBS) for 45 min per wash, in 100% ethanol at 4 °C overnight, and replace with fresh 100% ethanol for 45 min each. After slow critical point drying, samples were sputter-coated with a 4.5 nm thick gold/palladium alloy coating and imaged. 

### 2.8. Artifact Printability Testing

Bioinks were formulated as described in Section 2.2 with the addition of 0.01 mg fluorescent dye (46955, Sigma-Aldrich, St. Louis, MO, USA) per mL of bioink. Each syringe containing bioink was equilibrated at 17 °C for 10 min and then printed using the integrated tissue organ printer (ITOP) system, which has previously been described [15]. Printability was evaluated using a recently developed, bioink-specific artifact [19,20]. Briefly, four different structures were printed in triplicate, including a 5-layer tube, crosshatch, turn accuracy, and overhang collapse structure. Each print was conducted using a cylindrical nozzle with 330 µm inner diameter, a feed rate of 150 mm/min, a flowrate of 1.4 ± 0.05 mg/s, and a layer height of 0.42 mm. After printing, the artifact was weighed to confirm proper material deposition had occurred (400 ± 20 mg). 

Each artifact structure was photographed using a PowerShot SX730 HS camera (Canon, Tokyo, Japan). Images were taken in a dark room with a black background and UV light filtered at 365 nm (UV301D, LIGHTFE, Shenzhen, China) to enhance the contrast between printed bioink and background. Additionally, a ruler was included in the same plane as each structure for a pixel to mm conversion. Turn accuracy, crosshatch, and 5-layer tube structures were photographed from above at a distance of 5.5 cm. Overhang collapse and 5-layer tube structures were photographed from the side at a distance of 90 cm and a zoom of approximately 1.2 m to acquire the appropriate perspective. Images were analyzed using a custom MATLAB script (MathWorks, Natick, MA, USA). Select measurements are presented in Figure 3, and all artifact data can be found in Appendix A. More detailed descriptions and calculations for all measures can be found [19]. 

### 2.9. Statistical Analysis 

Data for each experimental group was expressed as the mean ± standard deviation (SD), and statistical significance determined using statistical analysis software (GraphPad Prism, Graphpad Software Inc., San Diego, CA, USA). Mixed models analysis of variance (ANOVA) was used to compare the results.

## 3. Results and Discussion

A graphical model describing the matrix components of fibrinogen hydrogel, decellularized skin ECM (dsECM), fibrinogen hydrogel supplemented with dsECM, and healthy human skin ECM is in Figure 1B. Normal human skin ECM is composed primarily of collagen I and collagen III, with added support from elastin, and the addition of glycoproteins and glycosaminoglycans [21]. The fibrinogen hydrogels typically used for bioprinting applications are commonly supplemented with gelatin and hyaluronic acid [22]. DsECM contains multiple ECM components, including collagen I and collagen III fragments, glycoproteins, and glycosaminoglycans [23]. The combination of fibrinogen hydrogel and dsECM in our study supports the structural integrity of the printed construct and provides normal skin ECM components. 

Human skin constructs made by encapsulating cells in fibrinogen hydrogel, either with or without dsECM, were matured in vitro over 15 days. H&E staining of the samples demonstrates improved cellularity over time in the fibrinogen + dsECM constructs compared with fibrinogen only (Figure 1C). The histological findings are strengthened by live/dead cell viability assays, which demonstrate stable viability in fibrinogen + dsECM and fibrinogen hydrogel only at day 8, with a significantly improved maintenance in viability at day 15 in the fibrinogen + dsECM constructs (73.8% ± 3.7 vs. 55.3% ± 2.8; *p* < 0.05; Figure 1D). Furthermore, the fibrinogen + dsECM constructs have greater structural stability than fibrinogen only controls. The cellularized fibrinogen + dsECM constructs maintain storage modulus over time, with significantly improved outcomes compared with the cellularized fibrinogen only controls after 15 days in culture (452.6 vs. 207.7; Figure 1E). Finally, the cellularized fibrinogen + dsECM constructs have improved storage modulus parameters after 15 days in culture, compared to cell-free fibrinogen + dsECM constructs, suggesting that cells in the constructs improve the mechanical strength of the skin constructs over time.

Next, we compared the physical properties of fibrinogen + dsECM to those of fibrinogen hydrogel only and human dsECM only. We found that the human skin ECM solution has a gelatinous form after pepsin digestion, and while neutralization is essential for cell viability, it causes phase separation (Appendix A) [12,24]. Rheological characterization confirms that the viscosity and G’ of neutralized human skin ECM is insufficient for bioprinting (Appendix A). Thus, we elected to use the neutralized human skin ECM as a supplement to fibrinogen hydrogel. The storage modulus (G’) temperature sweep demonstrates that the fibrinogen + dsECM has a high G’ at low temperatures, similar to fibrinogen only, and relatively high G’ at high temperatures, similar to dsECM only (Figure 2A). These findings suggest that the gel will better maintain structure during the printing process and culture in vitro and later when exposed to body temperature. Shear-thinning properties are optimal for hydrogel printability for extrusion bioprinting, with the gel acting more like a solid at a low shear rate (before extrusion and after extrusion), and more like a liquid at a high shear rate (during extrusion through the nozzle) [25]. A sweep of viscosity over shear rate demonstrates that the fibrinogen hydrogel supplemented with dsECM had viscous dsECM-like properties at low shear rates and liquid properties at high shear rates (Figure 2B). A sweep of viscosity over temperature demonstrates that the fibrinogen + dsECM has increased viscosity at low temperatures, similar to fibrinogen only hydrogel, and much better than dsECM only (Figure 2C). Taken together, the results of physical and structural characterization suggest that fibrinogen + dsECM has improved properties to support extrusion bioprinting compared with fibrinogen hydrogel only.

The effect of crosslinking on the stiffness of a hydrogel is also crucial in determining the physical characteristics for bioprinting applications. Mechanical strength is assessed by measuring the storage modulus pre and post-crosslinking with thrombin through a strain sweep test. The findings demonstrate an increase in the mechanical strength of fibrinogen hydrogel supplemented with dsECM (Figure 2D). The storage modulus of dsECM only has the highest natural storage modulus, but it drops significantly after the addition of a thrombin cross-linker over 1 h at room temperature (333.6 ± 34 vs. 51.1 ± 1.8; *p* < 0.05). As expected, the storage modulus of both the fibrinogen hydrogel and fibrinogen hydrogel + dsECM increase significantly after thrombin crosslinking, with the most considerable increase seen in fibrinogen hydrogel + dsECM (116.5 ± 4.4 vs. 780.3 ± 22.6; *p* < 0.05). Scanning electron microscopy (SEM) further demonstrates the structural composition of the hydrogels and normal decellularized human skin (Figure 2E). The dsECM hydrogel shows large swelling collagen bundles, while the fibrinogen hydrogel shows a network of small fibrin fibrils. The fibrinogen + 1% dsECM is made up of a composite of swelling bundles interlocked in a fibrin network. These findings are most similar to normal decellularized human skin, which shows both swelling collagen bundles and small-fiber networks. Together, these data suggest that the fibrinogen hydrogel + dsECM retains the physical properties of both the dsECM and fibrinogen hydrogel.

Next, we validated the improved physical properties of fibrinogen + dsECM using a printability artifact. Printability artifacts have been developed to test the effects of a wide range of factors on extrusion bioprinting, and are used to determine the suitability of a hydrogel for varying bioprinting applications [26]. A summary of all the artifact test findings by hydrogel type and representative images of artifact printability tests in triplicate by hydrogel type are in Figure 3E–F. Filament thickness, wall thickness, and crosshatch pore size are assessed for fibrinogen only hydrogel and fibrinogen + dsECM (Figure 3A–C). Fibrinogen + dsECM trend toward a thinner filament thickness (fibrinogen, 1.2 ± 0.1; fibrinogen + 1% dsECM, 1.1 ± 0.1; fibrinogen + 2% dsECM, 1.1 ± 0.1; *p* = 0.152) and wall thickness (Fibrinogen, 2.2 ± 0.3; Fibrinogen + 1% dsECM, 1.8 ± 0.2; Fibrinogen + 2% dsECM, 2.1 ± 0.2; *p* = 0.126) which correlates with a significant increase in crosshatch pore size, particularly in the 1% dsECM supplemented group (fibrinogen, 1.6 ± 0.3; fibrinogen + 1% dsECM, 2.6 ± 0.2; fibrinogen + 2% dsECM 2.2 ± 0.2; *p* < 0.01). Lattice printing patterns, with alternating crosshatch form, produce small micropores in printed constructs [17]. Further, microporous structures can improve diffusion and enhance the ingrowth of host blood vessels [18]. Finally, an overhang structure on the artifact demonstrates the filament strength of each hydrogel. The measured deflection from the overhang structure demonstrates a significant improvement in the fibrinogen hydrogel supplemented with 1% dsECM at the 16 mm overhang (fibrinogen, −3.0 ± 0.0; fibrinogen + 1% dsECM, 2.5 ± 0.2; *p* < 0.01; Figure 3D). Furthermore, fibrinogen hydrogel supplemented with 1% dsECM improves 16 mm overhang (66% success), while the fibrinogen hydrogel failed all attempts (0% success). The dsECM supplemented hydrogels demonstrate less deflection at 8 mm, 4 mm, 2 mm, and 1 mm than the fibrinogen only hydrogels, suggesting an overall improvement in shape fidelity in the dsECM supplemented hydrogels. While these data support 1% dsECM as the optimal concentration of dsECM as a supplement to fibrinogen hydrogel, future studies should incorporate concentrations greater than 2% ECM to determine if a true saturation effect is present.

## 4. Conclusions

In conclusion, we have shown the improved biological, physical, and printability properties of fibrinogen hydrogel supplemented with dsECM for skin bioprinting applications. The combination of fibrinogen hydrogel and dsECM provides the necessary ECM components to support the structural integrity and viability of cellularized bioprinted skin. Furthermore, assessment of the micro-architecture of the hydrogels with SEM demonstrates a composite of swelling bundles interlocked in a fibrin network, most similar to normal decellularized human skin. These findings correlate with improved rheological properties and shear thinning properties, incorporating the best of both the dsECM and fibrinogen hydrogels. Finally, fibrinogen hydrogel + dsECM demonstrates a significant improvement in the ability to print porous, crosshatch structures. These findings suggest that incorporating properties of dsECM hydrogel with fibrinogen hydrogel could improve in vivo integration of the bioprinted skin constructs, including neovascularization that will support healthy skin wound regeneration.

## Figures and Tables

**Figure 1 nanomaterials-10-01484-f001:**
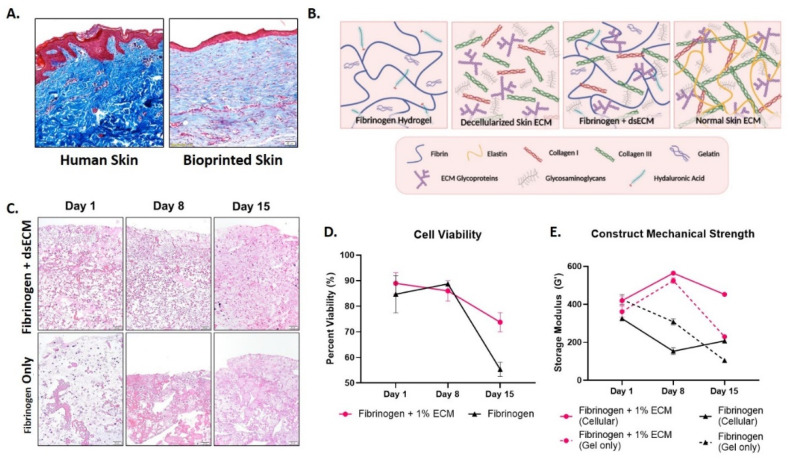
Improved biological characteristics of fibrinogen hydrogel supplemented with decellularized human skin extracellular matrix (dsECM). (**A**) Masson’s trichrome stained human skin compared with bioprinted skin bioprinted with a fibrinogen hydrogel (scale bars, 50 µm). (**B**) A graphical model describing the matrix components of fibrinogen hydrogel, decellularized skin ECM (dsECM), fibrinogen hydrogel supplemented with dsECM, and normal human skin ECM (created with BioRender.com). (**C**) H&E stained molded skin constructs with fibrinogen hydrogel supplemented with dsECM vs. fibrinogen only over 15 days (scale bars, 50 µm). (**D**) Live/dead cell viability assay demonstrates that viability was better maintained at day 15 in the fibrinogen hydrogel supplemented with dsECM. (**E**) Rheological assessment demonstrates improved mechanical strength of fibrinogen hydrogel supplemented with dsECM over 15 days in culture, with increased mechanical strength in cellularized constructs compared with gel only constructs.

**Figure 2 nanomaterials-10-01484-f002:**
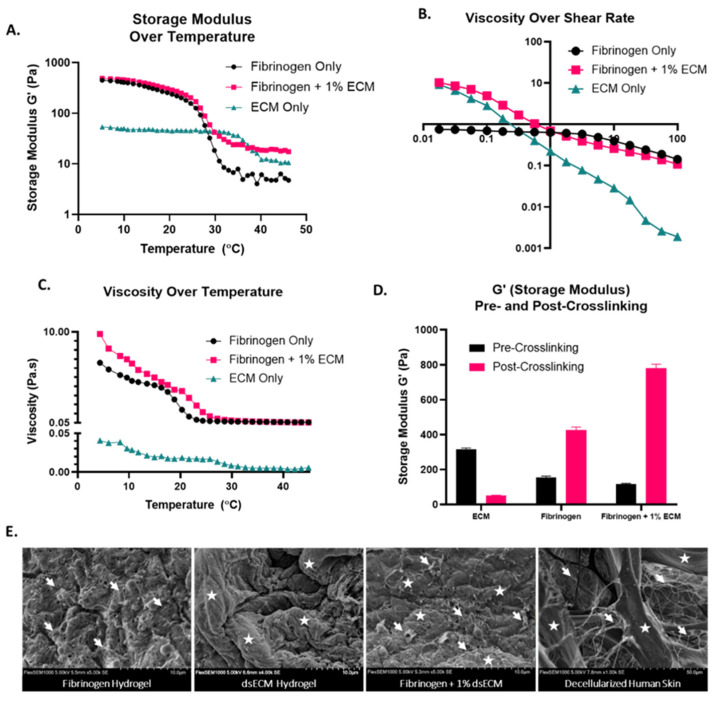
Improved physical characteristics of fibrinogen hydrogel supplemented with decellularized human skin extracellular matrix (dsECM). Rheological tests including (**A**) storage modulus (G’) temperature sweep, (**B**) viscosity over shear rate, and (**C**) viscosity temperature sweep demonstrate the shear-thinning properties and theoretical printability of the fibrinogen + dsECM hydrogel. (**D**) Storage modulus (G’) pre and post-crosslinking with thrombin. (**E**) Scanning electron microscopy (SEM) of each gel type compared to decellularized human skin; small fibrils (arrows, →) and swelling bundles (stars, ☆).

**Figure 3 nanomaterials-10-01484-f003:**
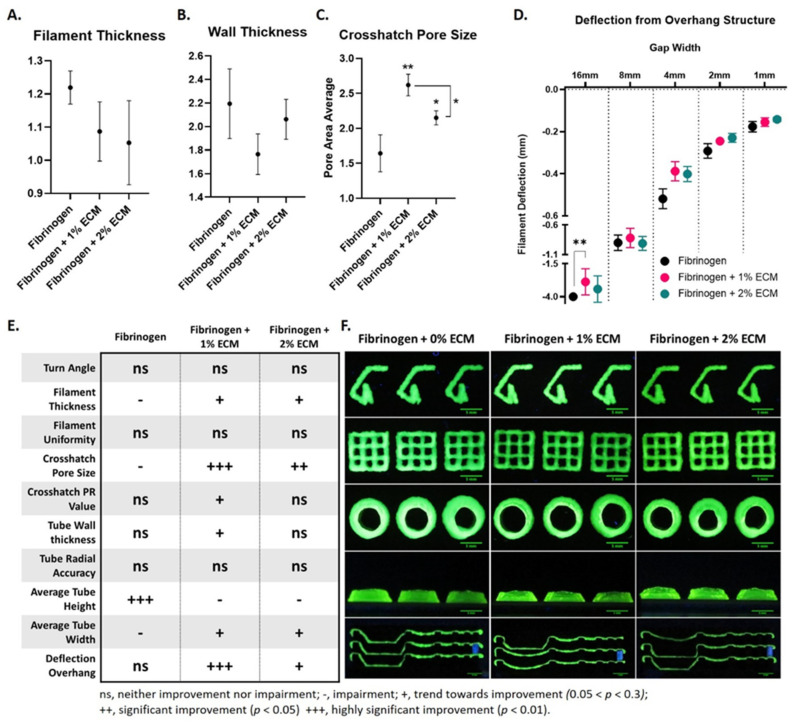
Improved extrusion bioprinting printability of fibrinogen hydrogel supplemented with decellularized human skin extracellular matrix (dsECM) assessed using a printability artifact. (**A**) Filament thickness, (**B**) wall thickness, (**C**) crosshatch pore size, and (**D**) deflection from the overhang structure were determined using a printability artifact. (**E**) Summary of artifact test findings by hydrogel type (ns, neither improvement nor impairment; −, trend towards impairment; +, trend towards improvement; ++, significant improvement, *p* > 0.05; +++, highly significant improvement, *p* > 0.01). (**F**) Representative images of artifact printability tests in triplicate by hydrogel type.

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
