# Peer review of "Decellularized Skin Extracellular Matrix (dsECM) Improves the Physical and Biological Properties of Fibrinogen Hydrogel for Skin Bioprinting Applications"

_nanomaterials, 2020, doi:10.3390/nano10081484_

Round 1

Reviewer 1 Report

In this article, the author evaluated the physical and biological characteristics of fibrinogen hydrogel supplemented with decellularized human skin-derived extracellular matrix, which could improve in vivo integration of the bioprinted skin constructs, so it may be used in healthy skin wound regeneration. It is better for them to further research the practice application of the fibrinogen hydrogel + dsECM. This work is promoted using scientifical method, and the experiment data supports the results completely. Therefore, this work is recommended for publication in nanomaterials after minor revision. The details are presented as the follow:

1. Title section, the author denoted the “Skin extracellular matrix” as “dsECM”, while in the text, “decellularized skin ECM (dsECM)” was denoted as “dsECM”, I suggested that the author should make them consistent.

2. Introduction section, I think the author should rewrite the introduce section and demonstrate more background of the research.

3. P4, Line 167, the number of “Results and Discussion” needs to be changed to “3”. P9, Line 285, the number of “Conclusions” needs to be changed to “4”

4. Results and Discussion session, P7, Line 217-218, “The human skin ECM solution has a gelatinous form after pepsin digestion, and while neutralization is essential for cell viability, it causes phase separation” The authors need to add some references to here.

5. Language and Format

The authors should re-read and correct some errors in English language, especially tense. Please check the references for formatting inconsistencies.

Author Response

Thank you for your comments. We have revised our manuscript entitled “Skin extracellular matrix (dsECM) improves the physical and biological properties of fibrinogen hydrogel for skin bioprinting applications,” according to the reviewers' comments using “tracked changes”. The revisions are as follows:

Comments from Reviewer 1:

  1. Title section, the author denoted the “Skin extracellular matrix” as “dsECM”, while in the text, “decellularized skin ECM (dsECM)” was denoted as “dsECM”, I suggested that the author should make them consistent.

Thank you, this has been corrected with the title changed to: “Decellularized skin extracellular matrix (dsECM) improves the physical and biological properties of fibrinogen hydrogel for skin bioprinting applications” (line 2)

  1. Introduction section, I think the author should rewrite the introduce section and demonstrate more background of the research.

We have revised the introduction and add two additional paragraphs describing 1) the contribution of ECM to tissue structure and function, and 2) the process of decellularization. We feel that these contributions have strengthened the introduction section. (lines 36-72)

  1. P4, Line 167, the number of “Results and Discussion” needs to be changed to “3”. P9, Line 285, the number of “Conclusions” needs to be changed to “4”

Thank you, this has been corrected.

  1. Results and Discussion session, P7, Line 217-218, “The human skin ECM solution has a gelatinous form after pepsin digestion, and while neutralization is essential for cell viability, it causes phase separation” The authors need to add some references to here.

This sentence has been revised to: We found that the human skin ECM solution has a gelatinous form after pepsin digestion, and while neutralization is essential for cell viability, it causes phase separation (Supplemental Figure 2A) [24-25].  (Line 239)

[24] Saldin, L. T., Cramer, M. C., Velankar, S. S., White, L. J., & Badylak, S. F. (2017). Extracellular matrix hydrogels from decellularized tissues: structure and function. Acta biomaterialia, 49, 1-15.

[25] Wolf, Matthew T., et al. "A hydrogel derived from the decellularized dermal extracellular matrix." Biomaterials 33.29 (2012): 7028-7038.

  1. Language and Format: The authors should re-read and correct some errors in the English language, especially tense. Please check the references for formatting inconsistencies.

Thank you, we have re-read and corrected tense errors and spelling errors throughout the manuscript, and followed this with Microsoft word “spell check” and “Grammarly” spelling and grammar check software. (See tracked changes)

We have also updated the references for consistency. We also reformatted the references to correct any inconsistencies. (See lines 347-420)

Reviewer 2 Report

The manuscript by Jorgensen et al. describes a complete physico-chemical characterization of a mixture of de cellullarized extracellular matrix and fibrinogen. This composition leads to a hydrogel with improved properties compared with the two components. The topic is of interest and the manuscript is well designed. The only aspect that the authors might want to discuss more in details is related to the effect of adding increasing percentages of ECM. While interesting effects have been shown upt to 2% ECM in fibrinogen, it might be interesting to see (or at least discuss) whether greater amounts of ECM lead to a saturation effect (especially concerning the properties described in Figure 3).

Author Response

Thank you for your comments. We have revised our manuscript entitled “Skin extracellular matrix (dsECM) improves the physical and biological properties of fibrinogen hydrogel for skin bioprinting applications,” according to the reviewers' comments using “tracked changes”. The revisions are as follows:

Comments from Reviewer 2:

  1. The only aspect that the authors might want to discuss more in details is related to the effect of adding increasing percentages of ECM. While interesting effects have been shown up to 2% ECM in fibrinogen, it might be interesting to see (or at least discuss) whether greater amounts of ECM lead to a saturation effect (especially concerning the properties described in Figure 3).

The following sentence has been added: “While these data support 1% dsECM as the optimal concentration of dsECM as a supplement to fibrinogen hydrogel, future studies should incorporate concentrations greater than 2% ECM to determine if a true saturation effect is present.” (lines 306-308)

Reviewer 3 Report

The paper by professor Atala and colleagues reports on the evaluation of physical and biological properties of fibrinogen hydrogel supplemented with decellularized human skin-derived extracellular matrix (dsECM). A comparative study with fibrinogen hydrogels only and dsECM hydrogels only was provided.

Cell viability and structural strength of bioprinted hybrid hydrogels were evaluated. A rheological characterization, including the study of i) pH effect on solubilized ECM; ii) flow property; iii) Thrombin-Fibrinogen crosslinking effect; iv) constructs mechanical strength, was carried out. The printability of fibrinogen hydrogels + dsECM was also evaluated, demonstrating remarkable improvements in the ability to print porous crosshatch structures. The rheological properties and the shear thinning properties of fibrinogen hydrogel supplemented with dsECM were shown to be improved, combining the positive characteristics of fibrinogen hydrogels and dsECM hydrogels. Scanning electron microscopy studies demonstrated a composite of swelling bundles interlinked in a fibrin network, similar to decellularized human skin.

The manuscript focuses on interesting and important topics, related with the development of new technologies for the treatment of full-thickness injuries. The results reported in the paper demonstrate improved biological, physical and printability properties of fibrinogen hydrogels + dsECM, therefore opening new spaces for potential applications in supporting healthy skin wound regeneration.

The manuscript is well written and the results are well discussed and clearly presented.

On the basis of all of these considerations, I recommend this paper for publication in Nanomaterials in the present form.

Author Response

Thank you for the recommendation to accept this manuscript.